# Nucleic-acid-base photofunctional cocrystal for information security and antimicrobial applications

Wenqing Xu[1,2,3,7], Guanheng Huang[1,7], Zhan Yang[4], Ziqi Deng[1], Chen Zhou[5], Jian-An Li[6], Ming-De Li [5] ✉, Tao Hu [2,3] ✉, Ben Zhong Tang [4] ✉ & David Lee Phillips [1] ✉

Cocrystal engineering is an efficient and simple strategy to construct functional materials, especially for the exploitation of novel and multifunctional materials. Herein, we report two kinds of nucleic-acid-base cocrystal systems that imitate the strong hydrogen bond interactions constructed in the form of complementary base pairing. The two cocrystals studied exhibit different colors of phosphorescence from their monomeric counterparts and show the feature of rare high-temperature phosphorescence. Mechanistic studies reveal that the strong hydrogen bond network stabilizes the triplet state and suppresses non-radiative transitions, resulting in phosphorescence even at 425 K. Moreover, the isolation effects of the hydrogen bond network regulate the interactions between the phosphor groups, realizing the manipulation from aggregation to single-molecule phosphorescence. Benefiting from the long-lived triplet state with a high quantum yield, the generation of reactive oxygen species by energy transfer is also available to utilize for some applications such as in photodynamic therapy and broad-spectrum microbicidal effects. In vitro experiments show that the cocrystals efficiently kill bacteria on a tooth surface and significantly help prevent dental caries. This work not only provides deep insight into the relationship of the structure-properties of cocrystal systems, but also facilitates the design of multifunctional cocrystal materials and enriches their potential applications.

Organic cocrystal materials can have numerous advantages such as facile preparation, low cost, adjustable morphology, and multifunctional characteristics and therefore have attracted considerable attention in photoconversion fields of research[1]. Photofunctional cocrystal materials with fascinating light harvesting properties are widely utilized in luminescent materials[2–9], water evaporation[10–12], organic photovoltaics[13,14], and photothermal imaging[15]. These crystalline single-phase materials are generally composed of two or more

[1]Department of Chemistry and State Key Laboratory of Synthetic Chemistry, The University of Hong Kong, Pokfulam Road, Hong Kong 999077, China. [2]State Key Laboratory of Oral Diseases & National Clinical Research Center for Oral Diseases, Sichuan University, Chengdu 610041 Sichuan, China. [3]Department of Preventive Dentistry, West China Hospital of Stomatology, Sichuan University, Chengdu 610041 Sichuan, China. [4]School of Science and Engineering, Shenzhen Institute of Aggregate Science and Technology, The Chinese University of Hong Kong, Shenzhen 518172 Guangdong, China. [5]Key Laboratory for Preparation and Application of Ordered Structural Materials of Guangdong Province, Department of Chemistry, Shantou University, Shantou 515031 Guangdong, China. [6]Sustainable Energy and Environment Thrust, The Hong Kong University of Science and Technology (Guangzhou), Nansha, Guangzhou 510000 Guangdong, China. [7]These authors contributed equally: Wenqing Xu, Guanheng Huang. ✉e-mail: mdli@stu.edu.cn; hutao@scu.edu.cn; tangbenz@cuhk.edu.cn; phillips@hku.hk

components through non-covalent interactions which show a more stabilizing lattice energy than the sum of their coformers[1]. The morphology structure and properties of cocrystals can be manipulated by selecting raw rational materials to regulate the interactions between the composites of the molecules. In addition, the cocrystals often exhibit some unpredicted chemicophysical properties that differ from the individual molecules due to the collaborative effects between the building block molecules, such as room temperature phosphorescence (RTP)[16,17], two-photon absorption[18], optical waveguides[19,20], etc. Therefore, cocrystal engineering not only emerges as a promising strategy for constructing multifunctional materials but also provides a platform to reveal the structure-function relationships at a molecular level.

To achieve the preparation of a photofuctional cocrystal, various strategies have been developed including employing charge transfer interactions[10,15,21–23], π-π interactions[24,25], halogen bonds[26,27], and hydrogen bonds[28–30]. Particularly in the field of life science, hydrogen bonding is an indispensable and fundamental non-covalent interaction[31–33]. Moreover, the hydrogen bond is a facile and reliable directional interaction to construct photofunctional cocrystals[1]. With notable strength and directionality, the hydrogen bond is the basis of some supramolecular cocrystals. Basically, the hydrogen bonding networks are composed of hydrogen donors and acceptors.

The base is a classic hydrogen bond-building unit, as one of the fundamental components of ribonucleic acid (RNA), with excellent programming properties, molecular recognition, and biocompatibility, which has become a research hotspot for functional materials in the field of biomedicine[34–38]. Due to their significance in life systems and some potential electronic applications, the photophysical and photochemical properties of bases have been extensively and deeply studied[39–42]. However, these previous studies of bases in the excited state have been mostly concentrated on the solution phase, while rarely researched in the condensed cocrystal state. Some bases show RTP properties in the solid state but the efficiency is poor because the short-lived triplet state is dissipated readily[43]. It is significant to enhance the RTP properties of the base because these materials exhibit huge potential in bioimaging, data storage, and anti-counterfeiting[44–50]. On one hand, forming strong hydrogen bonding networks can suppress the nonradiative transitions and stabilize the triplet state of the base to achieve a high performance of phosphorescence[51,52]. In the rigid hydrogen bond network, the phosphorescence molecules are fixed by hydrogen bonds, and the molecular motion is greatly restricted, resulting in the strong suppression of the non-radiative transition channel. Therefore, the most triplet state via radiative channel returns to the ground state leading to the significant enhancement of lifetime and quantum yield of phosphorescence[53–55]. On the other hand, this can also potentially enable effective energy transfer to molecular oxygen and form reactive oxygen species as a result of the long-lived and sufficient population of the triplet state species. With the generation of reactive oxygen species under light illumination, photofunctional cocrystals have potential applications in antimicrobial and anti-caries fields (Fig. 1). Therefore, through constructing hydrogen bonding networks, we expect to be able to exploit the multifunctional base-based photofunctional cocrystal materials and expand the fields of their application.

## Results and discussion

To validate our concept, we developed two co-assembled nucleic acid base photofunctional cocrystal materials constructed by robust hydrogen bonding networks. Uracil (U) was selected as a hydrogen acceptor and melamine (MA) or boric acid (B) was employed as a hydrogen donor to form the hydrogen bonding networks to construct the cocrystals of interest. Uracil was selected because of its multiple hydrogen bonding sites and room-temperature phosphorescence properties. The U shows weak room temperature phosphorescence properties due to the moderate spin orbital coupling and H-aggregated packing mode. Both melamine and boric acid also have abundant hydrogen bonding sites which are favored to construct high-strength hydrogen bonding networks. The process of preparing a cocrystal is low-cost and convenient. The three monomers are co-assembled by solvent evaporation to form two cocrystals (U-MA, U-B). The SEM images reveal that the two cocrystals exhibit completely different morphologies (Supplementary Fig. 1). U-MA is in the form of large flakes and U-B is in the form of very fine needles (Supplementary Fig. 2). The single crystal structure of U-MA demonstrates that U and

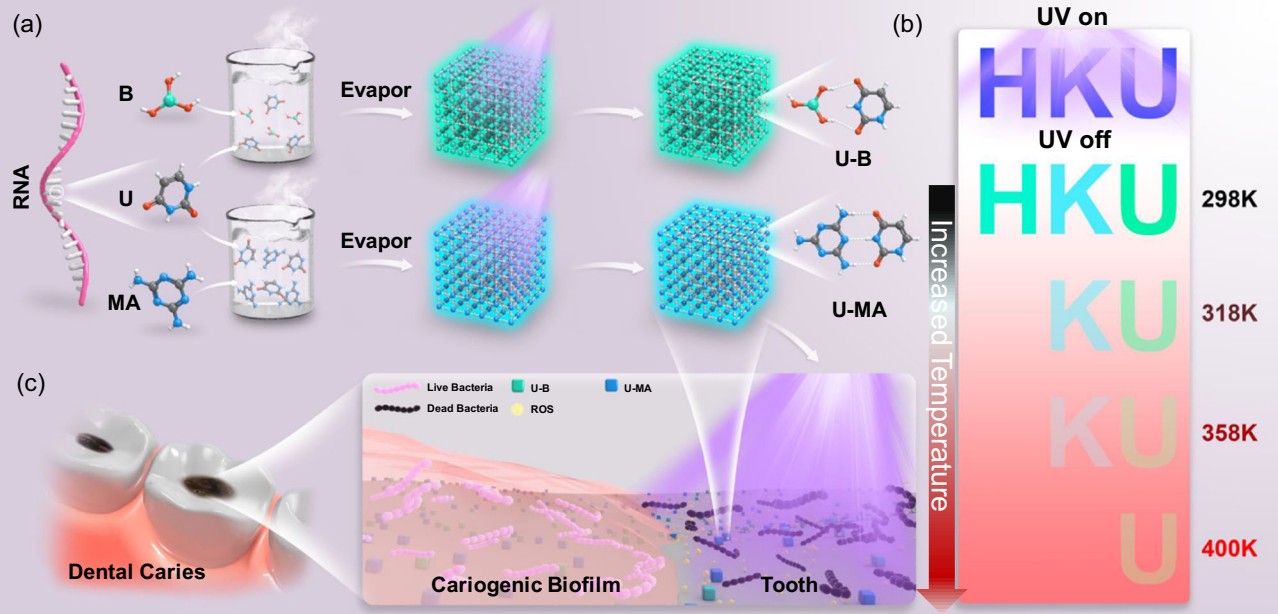

**Fig. 1 | Schematic illustration of nucleic-acid-base photofunctional cocrystal.** **a** The chemical structures of three monomers (U: uracil, MA: melamine, and B: boric acid) and two cocrystals (U-MA and U-B). **b** The temperature responsive phosphorescence utilized in multi-level information storage. **c** Potential antimicrobial application of U-MA and U-B for dental caries prevention under 365 nm light.

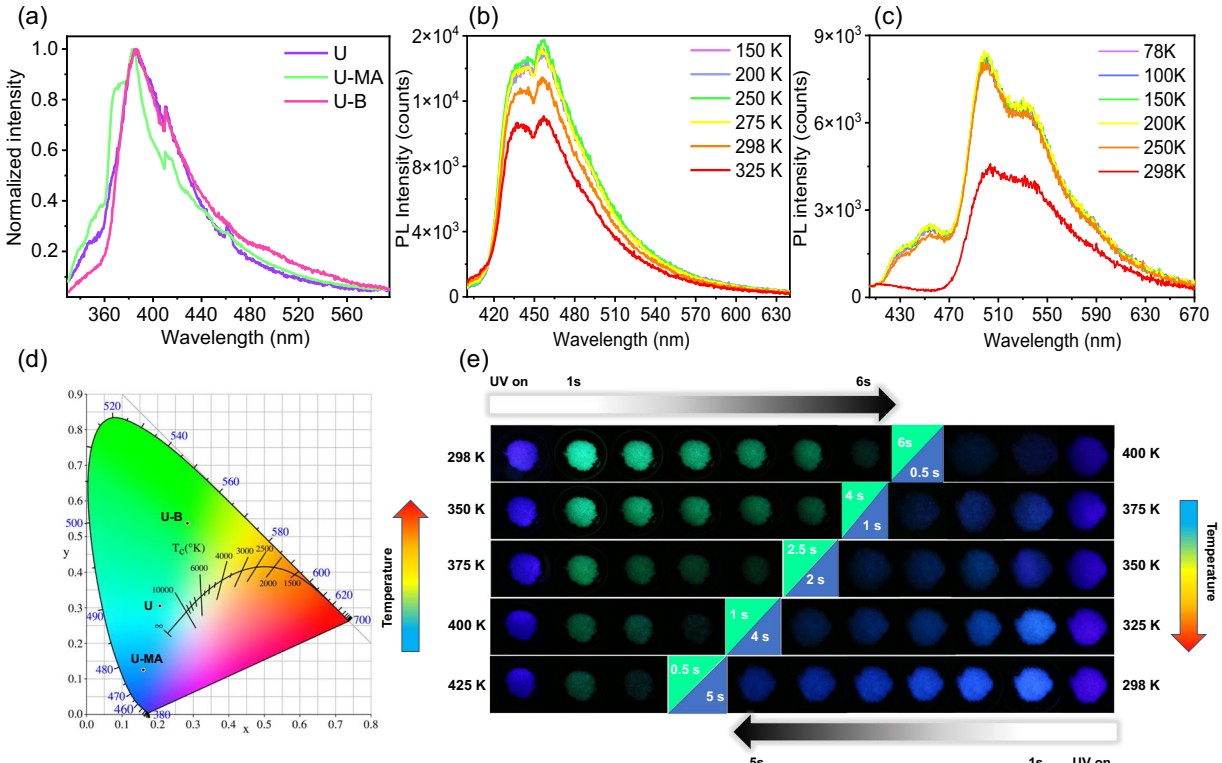

**Fig. 2 | Photophysical characterization and afterglow properties of cocrystals.**
**a** Steady-state photoluminescence spectra upon excitation at 365 nm under
ambient condition. **b** Phosphorescence spectra of the U-MA cocrystal under
vacuum conditions at different temperatures. **c** Phosphorescence spectra of the
U-B cocrystal under vacuum conditions at different temperatures. **d** The CIE
coordinates of the afterglow of U, U-MA, and U-B under 365 nm excitation.
**e** Afterglow color and duration of U-MA and U-B at different temperatures.

MA form hydrogen networks by multiple intermolecular hydrogen
bond interactions (Supplementary Fig. 3). Since the cocrystal of U-B is
very fragile and tiny, we failed to characterize the single crystal
structure of U-B. We simulated the structure of U-B by employing
calculations (Supplementary Fig. 4). The multiple strong hydrogen
interaction of U-MA and U-B resulting in the arrangement of molecules
tends to a planar configuration, and each U molecule is locked by
multiple hydrogen bonds in the plane. These results indicate that
U-MA and U-B cocrystals exhibit robust hydrogen bond network.
Ultralong bright afterglow of U-MA and U-B lasting for more than 6 s is
observed by the naked eye after removing the light source. Unex-
pectedly, the U-MA cocrystal exhibits ultralong blue phosphorescence
which is an obvious blue-shift compared with the U individual mole-
cule emission. In addition, these two cocrystals retain afterglow
properties even at 425 K. The blue-shift of the phosphorescence and
the high-temperature resistant properties demonstrate that the
hydrogen bonding network imbues some undiscovered features to
these cocrystals.

### Photophysical properties of cocrystals

We next investigated the photophysical properties of the U and the
two cocrystals under ambient conditions. Both UV-Vis and steady-state
photoluminescence spectra of the two cocrystals show no new peaks
and display superimposed spectra of two constructed monomers
(Fig. 2a and Supplementary Fig. 5). These results indicate that no
charge transfer occurs between the two monomers. From delayed
photoluminescence spectra, U-MA shows an afterglow emission at
440 nm while U-B is at 540 nm (Supplementary Fig. 6). The afterglow
emission of U overlaps the emission of the two U-comprised cocrys-
tals. Also, U shows different kinetics at 450 nm and 500 nm, indicating
that the afterglow emissions are from different decay channels

(Supplementary Fig. 7). The phosphorescence nature of the two
cocrystals and U is confirmed by temperature-dependent spectra and
kinetics (Fig. 2b, c and Supplementary Figs. 8–11). CIE coordinates of
the afterglow display that U-MA shows blue phosphorescence, while
U-B is green and U is cyan (Fig. 2d). From the comparison of kinetic
decay, two cocrystals show significant enhancement in lifetime (Sup-
plementary Fig. 12) The lifetime of U is only 76 ms at 500 nm under
ambient conditions, while the lifetime of U-MA at 450 nm and U-B at
500 nm is 530 ms and 663 ms, which increase by nearly 10 times
compared to U. The phosphorescence quantum yield of U, U-MA, and
U-B are 3%, 21%, and 8.2% respectively. These two cocrystals exhibit
obvious enhancement of phosphorescent quantum yield compared
with the U monomer (Supplementary Fig. 13). To sum up, the above
results prove that forming the hydrogen bond network greatly
enhances the phosphorescence performance, including prolonging
the lifetime and increasing the quantum yield. Additionally, these two
cocrystals still exhibit afterglow properties from room temperature
and even up to 425 K (Fig. 2e). The cocrystal structure not only
improves the lifetime and quantum yield of phosphorescence, but also
enables these two cocrystals to have heat resistance that ordinary
phosphorescent materials do not have (Supplementary Fig. 14).
Notably, robust intermolecular interactions cause a bathochromic
shift in luminescence, especially in the solid state. However, compared
with the phosphorescence of U, U-MA exhibits a significant blue shift,
but U-B does not have this feature.

### Mechanistic investigations of room temperature phosphores-
cence of cocrystal

To shed light on the origin of blue phosphorescence and reveal the
photophysical processes of the two cocrystals, ultrafast femtosecond
transient absorption (fs-TA) was employed to investigate the

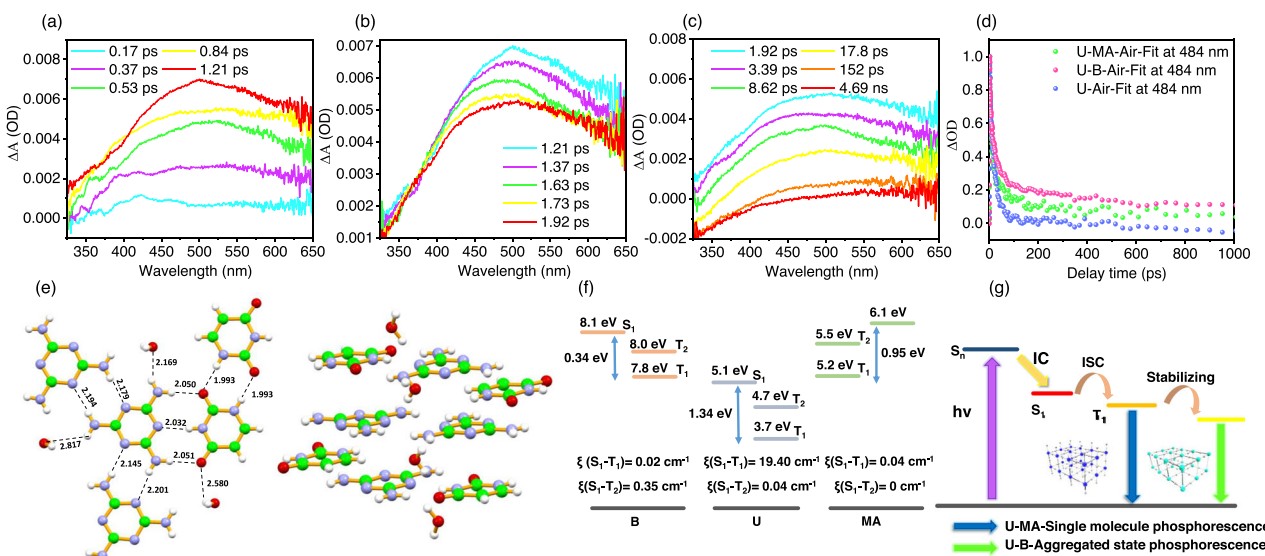

**Fig. 3 | Mechanistic investigations of RTP properties of cocrystals. a** fs-TA spectra of U·MA film under 290 nm excitation from 0.17 to 1.21 ps. **b** from 1.21 to 1.92 ps. **c** from 1.92 ps to 4.69 ns. **d** Kinetic fitting for U, U·MA, and U·B at 484 nm under excitation at 290 nm. **e** Single crystal structure of U·MA. **f** Energy level diagram and spin orbital coupling matrix element of U, U·MA, and U·B. **g** Proposed mechanism of phosphorescence of U·MA and U·B.

evolutionary pathways in the excited state. The 290 nm laser was selected as the pump light, and under this circumstance, the U is excited but MA is not excited significantly (Supplementary Fig. 5). After excitation, U·MA undergoes an internal conversion process to generate the lowest energy excited state species (Fig. 3a). Afterward, the absorption band blue shifted from 1.21 ps to 1.92 ps (Fig. 3b). Finally, the transient species decays gradually (Fig. 3c). The fs-TA spectrum of U·MA is very close to that previously reported for the spectral evolution of U in a dilute solution[56]. However, from the fs-TA spectra, only the generation of the lowest energy state and the decay process for U and U·B can be observed (Supplementary Figs. 15 and 16). Compared with a dilute solution, the transient absorption peak of U undergoes a bathochromic shift due to the strong intermolecular interactions in the crystalline state. From the kinetics fitting, U·B and U·MA show slower decay rates than U does which is consistent with the lifetimes being phosphorescence as mentioned above (Fig. 3d). The results of the fs-TA experiments indicate that the U·MA exhibits a single molecular photophysical process of U, while U·B exhibits similar properties to the aggregated U system. In addition, the low-temperature phosphorescence spectrum shows that U exhibits blue phosphorescence at 450 nm in a dilute solution (Supplementary Fig. 17). The phosphorescence spectrum of U·MA not only shows good agreement with the phosphorescence spectrum of U dilute solution, but also with the simulated spectra of U monomer, demonstrating that the blue phosphorescence of U·MA originates from single-molecule phosphorescence (Supplementary Fig. 18). However, the U·B cocrystal exhibits an aggregated state phosphorescence property.

This analysis of the single crystal structure demonstrates that U has a face-to-face parallel arrangement with an H-aggregation character through intensive π-π interactions (Supplementary Fig. 19). This packing mode configuration is beneficial to stabilize the triplet excitons to induce room temperature phosphorescence. Moreover, this H-aggregate structure can form a $T_1^*$ with lower energy than $T_1$, resulting in aggregated phosphorescence[57,58]. As a result, the phosphorescence of the aggregated state will be red-shifted compared with the phosphorescence of the single molecules. Therefore, U simultaneously exhibit single-molecule and aggregated phosphorescence which is consistent with the phosphorescence spectrum and kinetics result mentioned above (Supplementary Figs. 7 and 9). Interestingly,

the single crystal of U·MA shows that the triplet hydrogen bonding between U and MA is like the form of the complementary pairing of DNA, resulting in very strong intermolecular interactions (Fig. 3e). From the packing mode of U·MA, U monomers and MA form strong hydrogen bonding interactions in the same layer. In different layers, U monomers are completely staggered leading to no π-π interactions between the phosphorophores. Due to the hydrogen-bonding network of the cocrystal, each U is individually isolated, which is similar to the single-molecule state in a dilute solution. All of the phosphorophores are locked and isolated from each other by the hydrogen bonding networks. Confining the isolated chromophores by ionic bonding was achieved to realize the single molecular blue phosphorescence[59]. Hence, even in strongly interacting aggregated states, single-molecule blue phosphorescence of U·MA can be exhibited. Owing to the small molecular structure of boric acid, although it can form very strong hydrogen bonds with U, it is not capable of completely isolating the phosphorophores. The interactions and π-π stacking between U still exist, resulting in U·B exhibiting aggregated phosphorescence. In addition, the results of the calculations show that the energy gap of $T_1$ between U and MA or B is 1.5 eV and 4.1 eV, respectively (Fig. 3f). The process of triplet-to-triplet energy transfer from MA and B to U cannot occur because of the large energy gap of the triplet state. Additionally, the molecular orbital calculation and steady-state absorption and emission spectra indicate that no electron transfer occurs between the two monomers (Fig. 2a and Supplementary Figs. 5 and 20–23). Therefore, although MA and B provide very strong hydrogen bonding interactions, they do not participate in the generation of the excited state of U. Generally, single-molecule phosphorescence can be achieved by doping, and the guest is doped into the host with a trace dose[51,60]. However, the host-guest system suffers from phase separation, causing a decrease in its efficiency. Also, the excited state processes are very complicated, and the structural information of the doping system is not known which increases the difficulty for mechanistic research. The U·MA and U·B cocrystals formed by the strong hydrogen bonding network can not only reveal the mechanisms of the phosphorescence at the molecular level but also realize the regulation from aggregated phosphorescence to single-molecule phosphorescence that allows manipulating the afterglow color to range from green-cyan-blue.

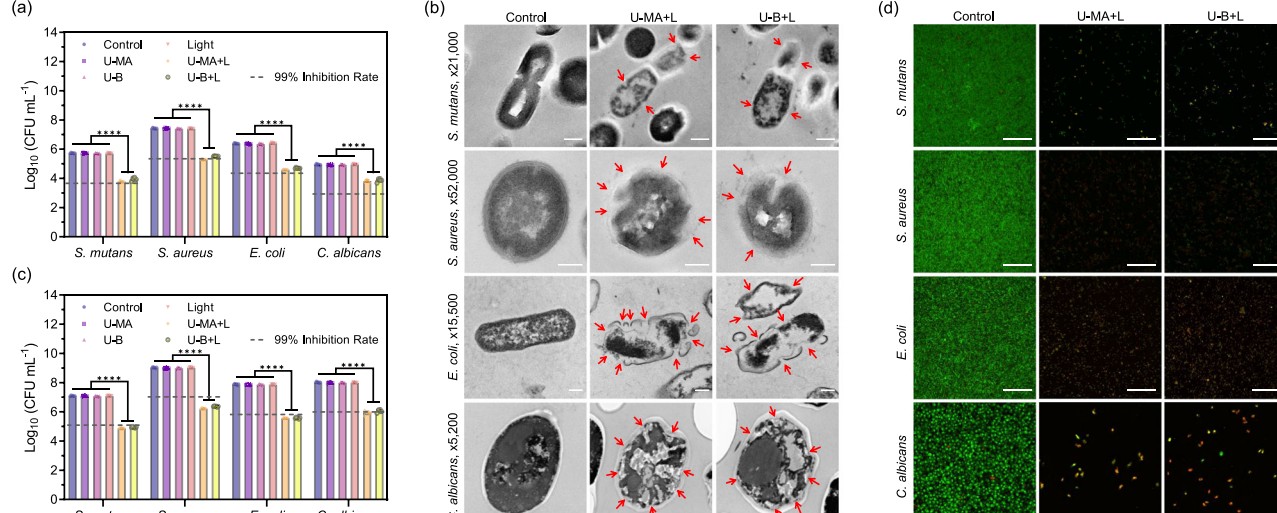

**Fig. 4 | Antimicrobial effect of PDT mediated by U-MA and U-B. a** Photodynamic killing of planktonic *S. mutans*, *S. aureus*, *E. coli*, and *C. albicans*. Error bars = Standard Deviation (*n* = 6 biologically independent samples). One-way ANOVA was performed followed by Tukey's multiple comparisons. ****$p < 0.0001$. **b** Morphological changes in microbes treated with or without U-MA or U-B mediated PDT by TEM. The arrows indicate the loss of typical cell shapes of the treated microorganisms. The scale bars are 200 nm. **c** Photodynamic inhibition of the formation of *S. mutans*, *S. aureus*, *E. coli*, and *C. albicans* biofilm. Microorganisms were cultured for 24 h after treatment. Error bars = Standard Deviation (*n* = 6 biologically independent samples). One-way ANOVA was performed followed by Tukey's multiple comparisons. ****$p < 0.0001$. **d** Biofilm viability detected by CLSM. Microorganisms were cultured for 24 h after treatment. The scale bars are 50 μm. L Light.

## Photodynamic killing of planktonic microorganisms

Due to the stabilizing effect of the hydrogen-bonding network, photofunctional cocrystals can efficiently generate long-lived triplet states. Long-lived triplet states are capable of energy transfer with ambient molecular oxygen to generate reactive oxygen species which can be applied in photodynamic therapy (PDT) applications. The reactive oxygen species can be utilized in PDT fields including antimicrobial[61] and antitumor[62] applications. In addition, since light for oral applications only needs to directly irradiate the teeth and does not need to penetrate tissue, ultraviolet light and blue light are widely used for this application. Therefore, the photofunctional cocrystal has potential applications in dentistry including dental caries treatment. A broad range of microorganisms, including *Streptococcus mutans* (*S. mutans*), *Staphylococcus aureus* (*S. aureus*), *Escherichia coli* (*E. coli*), and *Candida albicans* (*C. albicans*), were used to demonstrate the antimicrobial ability of U-MA and U-B. *S. mutans* is the primary pathogenic bacteria responsible for dental caries and plays a crucial role in caries progression[63]. *S. aureus*, *E. coli*, and *C. albicans* are among the most common types of Gram-positive bacteria, Gram-negative bacteria, and fungi, respectively. The ability of U-MA and U-B for photodynamic planktonic sterilization was assessed by bacterial plate counts. The results indicate that the survival rate of microbes did not differ significantly between the control group and the groups that underwent light irradiation, U-MA, or U-B treatment. However, U-MA and U-B exhibited strong photodynamic antimicrobial activity against all tested planktonic microorganisms (Fig. 4a, Supplementary Table 1, and Supplementary Figs. 24–27), especially *S. aureus*, with a survival rate of only 0.77% after exposure to light. This result demonstrates that these two cocrystals exhibit antimicrobial properties under photoexcitation.

To investigate the microbicidal mechanism(s) of U-MA and U-B under light, we utilized a Singlet Oxygen Sensor Green (SOSG) probe to detect reactive oxygen species. The increased intensity of fluorescence observed at 525 nm as the irradiation time increased indicated that the reactive oxygen species was singlet oxygen (Supplementary Fig. 28). Additionally, Transmission Electron Microscopy (TEM) was utilized to visualize the structural and morphological changes in the microbes following treatment (Fig. 4b). The control groups of all four microorganisms exhibited normal shape and intact cell membranes. In contrast, treatment with light and U-MA or U-B resulted in the loss of typical cell shape and disturbance of the cell walls and membranes in the treated microorganisms. Thus, the remarkable microbicidal activity of U-MA and U-B can be attributed to the photodynamic effect.

## Photodynamically inhibited biofilm formation

Based on the above-mentioned results, it can be inferred that U-MA and U-B possess strong antimicrobial properties against planktonic microorganisms, which may significantly impede biofilm formation. In the absence of either light, U-MA, or U-B, the formation of biofilms remained largely unaffected. However, when U-MA and U-B were combined with light, they exhibited remarkable efficacy in inhibiting biofilm formation (Supplementary Figs. 29–32), particularly against *S. aureus*, which showed the lowest survival rate, about 0.16%. Confocal Laser Scanning Microscopy (CLSM) was utilized to evaluate the effects of U-MA and U-B on biofilm formation inhibition. SYTO 9/propidium iodide (PI) kit was used to stain the microbes, with live and dead cells exhibiting green and red fluorescence, respectively. The control group that was exposed to light or treated with U-MA or U-B showed intense green fluorescence of SYTO 9 and minimal red fluorescence of PI, indicating that a significant number of microorganisms survived, multiplied, and formed biofilms (Supplementary Figs. 29–32). After treatment with U-MA or U-B mediated PDT, the bright green fluorescence of SYTO 9 was sporadic, and red fluorescence of PI was observed, indicating a small number of live microorganisms and significant inhibition of biofilm formation. In addition, the biocompatibility experiment using human gingival fibroblast (HGF) cells, which are the main component of the human gingiva, demonstrated the low cytotoxicity of U-MA and U-B (Supplementary Fig. 33). Therefore, U-MA or U-B exhibited robust photodynamic inhibition of biofilm formation against a wide range of microorganisms without causing cell damage.

## Application of the multifunctional cocrystal

On one hand, the unique high-temperature resistance and the multiple colors of the phosphorescence make these systems attractive for applications involving multi-level information security. As shown in

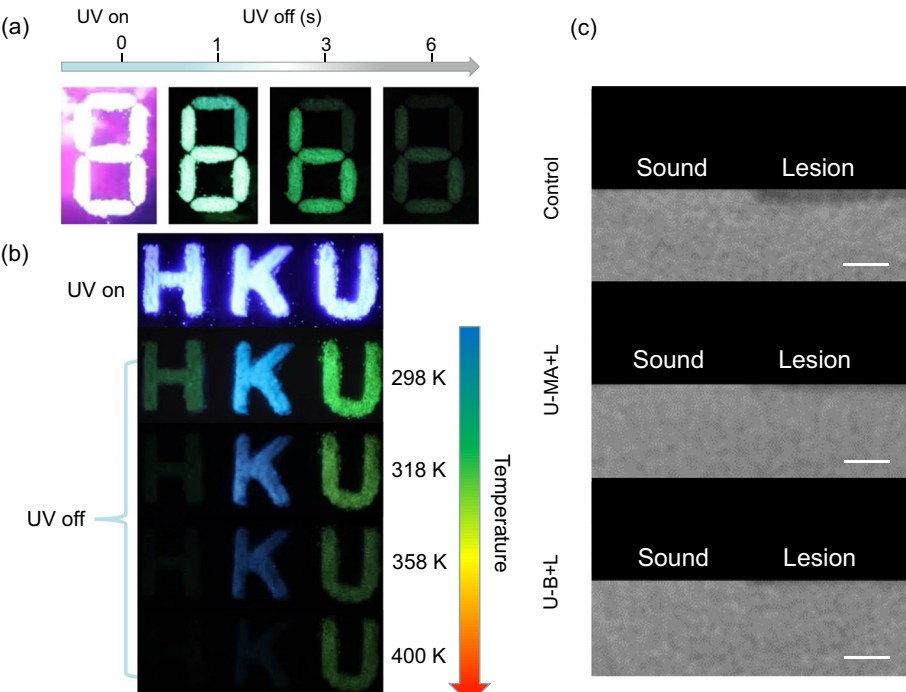

**Fig. 5 | The application of cocrystal materials. a** Photographs of the pattern "8" before and after removing UV irradiation. Pattern "6" is composed of U-B and the remaining part is composed of uracil. **b** Photographs of the pattern of "HKU" before and after removing UV irradiation. Alphabet H is composed of uracil, alphabet K is composed of U-MA and alphabet U is composed of U-B. **c** Micro-CT images of dentin with different treatments. The scale bars are 200 µm. L Light.

Fig. 5a, the pattern of "8" is composed of U and U-B. After removing the UV light, the pattern transforms from "8" to "6" realizing the time-gate information storage. Moreover, due to the high-temperature-resistant phosphorescence of the cocrystal, we fabricated the temperature-responsive information storage and anti-counterfeiting by utilizing the photofunctional cocrystal. The patterns of HKU are composed of U, U-MA, and U-B respectively. After erasing the UV light, visible multicolor afterglows were observed under the ambient conditions. By increasing the temperature from 298 K to 400 K, different information is observed in different ranges of temperature. In addition, two cocrystals exhibit excellent stability over temperature cycling (Supplementary Figs. 34 and 35). The temperature-responsive phosphorescence of photofunctional cocrystals demonstrates their potential applications in smart materials and muti-level information storage. On the other hand, since the photofunctional cocrystal exhibits a superior anti-microbial effect under light irradiation, which can kill the cariogenic microorganism *S. mutans*, it has the potential to prevent dental caries. The potential of U-MA and U-B-mediated PDT for anti-caries applications was assessed using bovine dentin blocks. To simulate the development of dental caries, dentin blocks were subjected to U-MA or U-B and light therapy after the addition of *S. mutans*, followed by anaerobic incubation at 37 °C for 72 h. The colony count results revealed that the number of bacteria on the dentin blocks treated with light and U-MA or U-B was only 0.11% of that in the control group (Supplementary Fig. 36). Micro-computed tomography (Micro-CT) was employed to analyze the degree of caries. The area without transmission shadow (white in Fig. 5c) is intact teeth, whereas the area with transmission shadow (black in Fig. 5c) is eroded by caries. Compared to the control group or dentin blocks treated with light or U-MA or U-B alone, dentin blocks treated with light plus U-MA or U-B exhibited reduced erosion (Fig. 5c). Collectively, these findings suggest that U-MA and U-B mediated PDT exhibits potential anti-caries effects.

In conclusion, we have realized two kinds of nucleic-acid-base photofunctional cocrystals by employing strong hydrogen bonding networks constructed in the form of complementary base pairing. The triplet state of the base can be stabilized by the strong hydrogen-bonding network, not only leading to improved phosphorescence performance (higher quantum yield and longer lifetime) but also realizing a high-temperature phosphorescence character. Notably, the unique packing mode formed by U-MA realizes single-molecule blue phosphorescence in crystalline form due to the phosphorophores being isolated from each other. Through the co-assemble strategy, the regulation of aggregated phosphorescence to single-molecule phosphorescence has been realized, making the afterglow color change from green-cyan-blue. Since the cocrystals efficiently generate a long-lived triplet state under light, singlet oxygen can be efficiently generated by energy transfer with the triplet state. Based on this photodynamic effect, photofunctional cocrystals can be used as antimicrobial agents. The in vitro experiments showed that the cocrystals exhibit a spectrum of antimicrobial ability under light. In addition, the cocrystals can kill cariogenic bacteria on the tooth surface under light, reduce tooth erosion, and achieve an anti-caries effect which has potential application in dentistry. Therefore, this co-assembly strategy via base-pairing hydrogen bonding may pave the way for the future development of nucleic acid-based materials for applications in display, data encryption, and antimicrobial applications.

## Methods

### Sample preparation
U (Uracil, 99%), MA (melamine, 99%), and B (boric acid, 99%) were purchased from J&K. All other solvents and reagents were purchased with analytical grade and used without further purification. For the preparation of cocrystals of U-MA, and U-B, the sample powder of U and MA or B (at a molar ratio of 1:1) was in a mixture of water and methanol at a molar ratio of 1:1. With the evaporation of solvent at room temperature, the cocrystal can be obtained.

### UV-vis absorption experiments
The UV-Vis absorption spectra of samples were recorded by a spectrometer (PE Lambda950) from the PERKINELMER company.

## Photoluminescence (PL) experiments

The PL spectra were recorded by a spectrometer (FLS980) from Edinburgh Instruments. Time-resolved PL decay kinetics and PL spectra were recorded in different temperatures by the Edinburgh company.

## Femtosecond transient absorption (fs-TA) experiments

Fs-TA measurements were performed with an apparatus and methods detailed previously and only a brief description is provided here[64]. The fs-TA measurements were done by using a femtosecond regenerative amplified Ti: sapphire laser system in which the amplifier was seeded with the 80 fs laser pulses from an oscillator laser system. The laser probe pulse was produced with ~5% of the amplified 800 nm laser pulses to generate a white-light continuum (350−800 nm) in a $CaF_2$ crystal and then this probe beam was split into two parts before traversing the sample. One probe laser beam went through the sample while the other probe laser beam went to the reference spectrometer in order to monitor the fluctuations in the probe beam intensity. After that, 290 nm was selected as the excitation light, and the excitation power was 0.2 mW. The single-wavelength kinetics fitting was used based on equation 1. The $\tau$ was referred to as delay time, and $\tau_0$ means zero time. The $\tau_p$, and A represented the instrument response value and the proportion of species respectively.

$$S(t) = e^{-\left(\frac{t-t_0}{t_p}\right)^2} * \sum_i A_i e^{-\frac{t-t_0}{t_i}}$$

## Structural characterization

The structure of the single crystals of U, U-MA were given by a Rigaku OD (Enhance Cu X-ray source, K$\alpha$, $\lambda$ = 1.54184 Å) using a CCD Plate (XtaLAB Pro: Kappa single) at 100 K. The structure of the crystal was solved by the direct method in Olex2.

## Theoretical calculations

The theoretical calculations were carried out in Gaussian 16 software packages[65]. CAM-B3LYP and 6−31 g(d) were selected as the functional and basis set to perform the DFT and ORCA calculations[66]. The structures of the cocrystals were shown in VMD software[67].

## Photodynamic killing of planktonic microorganisms

*S. mutans*, *S. aureus*, *E. coli*, and *C. albicans* were cultured as Supplementary Table 2 mentions respectively. For the minimum inhibitory concentration (MIC) test, serial two-fold dilution from 160 to 0.16 μM U-MA or U-B was added and the 365 nm 10 W UV light was exposed for 15 min. The distance between the bacteria and the UV lamp should be kept at approximately 50 cm to minimize the impact of the sterilizing effect of the UV lamp on the experiment. The Optical Density (OD) value was determined at 660 nm (bacteria) or 520 nm (fungus) by the microplate reader both before and after the microorganisms were cultivated for 24 h. MIC values were assessed from the wells that had the same OD value before and after the incubation and did not exhibit visible microbial growth. To guarantee uniformity, the experiment was conducted three times. For the photodynamic killing of the planktonic microorganisms, 40 μM U-MA or U-B was added and the 365 nm 10 W UV light was exposed for 15 min. The treated solution was serially diluted tenfold and inoculated on corresponding agar plates respectively and cultured. After 48 h, the number of colonies was counted. The formula used to determine the survival rate was $C/C_0 \times 100\%$, where C is the CFU of the sample that was exposed to light and/or U-MA or U-B, and $C_0$ is the untreated control group. To guarantee uniformity, the experiment was conducted with six independent samples. For microbial morphology assessment, the treated microorganisms were collected by centrifugation and fixed in 2.5% glutaraldehyde. The semi-thin slices of microbes were enclosed in grids and observed by the TEM (Philips CM100).

## Photodynamically inhibited biofilm formation

The above-mentioned light treatment and 40 μL U-MA or U-B treatments were carried out, and then cultured respectively for 24 h. The biofilm was collected and the number of colonies was counted and assessed as stated above. To analyze the biofilm viability, the treated microorganisms will be placed in a confocal dish and incubated for 24 h. Biofilms were evaluated under CLSM (OLYMPUS, Japan) after staining with SYTO 9/PI kit for 15 min. To guarantee uniformity, the experiment was conducted with six independent samples.

## Biocompatibility experiment

The HGF was cultured in Dulbecco's modified Eagle's medium (DMEM; Gibco, UK) supplemented with 10% fetal bovine serum (FBS; Gibco, UK) and 1 % penicillin–streptomycin solution (Invitrogen, UK). HGF was seeded on a 96-well plate at a concentration of $10^4$ per well and cultured for 24 h. The cells were then cultivated for another 24 h while being exposed to various U-MA or U-B concentrations. The CCK-8 kit (Dojindo, Japan) was used to assess cell viability. To guarantee uniformity, the experiment was conducted three times.

## Inhibition of dental caries in vitro

Sound bovine incisors were cut into dentin blocks and polished with 1200 grid micro-fine sandpaper under water cooling. For sanitization, these dentin blocks were autoclaved at 121 °C. The dentin blocks were infused with $10^8$ cells/mL *S. mutans* culture in BHI broth containing 5% sucrose and then exposed to light and/or U-MA or U-B treatment. The dentin blocks were cultured anaerobically at 37 °C for 72 h before being vortex-shaken to collect the bacteria. As mentioned above, the number of colonies was counted and evaluated. Micro-CT was used to detect the degree of caries in dentin blocks.

## Statistical analysis

Statistical differences were assessed using one-way ANOVA and Tukey's multiple comparisons with GraphPad Prism v.9.5, and the significance was declared when $p < 0.05$. No statistical method was used to predetermine the sample size. The reproducibility of findings was ensured by conducting multiple independent samples or experiments.

## Reporting summary

Further information on research design is available in the Nature Portfolio Reporting Summary linked to this article.

# Data availability

The data generated in this study are provided in the paper, the Supplementary Information and the Source Data file. Source data are provided with this paper.

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

## Acknowledgements

This project is financially supported by the Shenzhen Key Laboratory of Functional Aggregate Materials (ZDSYS20211021111400001, B.Z.T.), the Science Technology Innovation Commission of Shenzhen Municipality (KQTD20210811090142053, JCYJ20220818103007014, B.Z.T.). This work was supported by grants from the Hong Kong Research Grants Council (GRF 17302419, GRF 17316922, D.L.P.), Major Program of Guangdong Basic and Applied Research (2019B030302009, D.L.P.), Guangdong-Hong Kong-Macao Joint Laboratory of Optoelectronic and Magnetic Functional Materials (2019B121205002, D.L.P.) and Key-Area Research and Development Program of Guangdong Province (2020B0101370003, D.L.P.).

## Author contributions

W.X. and G.H. contributed equally to this work. W.X., G.H., T.H., B.Z.T., and D.L.P. conceived the study. G.H. synthesized cocrystals and performed the mechanism and application research of room temperature phosphorescence. W.X. performed the experiments on photodynamic antibacterial and potential dental applications. G.H., Z.Y., C.Z., and J.A.L. perform the photophysical characterization of cocrystal materials. Z.D. conducted the theoretical calculation. W.X., G.H., M.D.L., T.H., B.Z.T., and D.L.P. discussed the results and edited the manuscript.

## Competing interests

The authors declare no competing interests.
