## [Peer review file · Nature Communications]

Nucleic-Acid-Base Photofunctional Cocrystal for Information Security and Antimicrobial ApplicationsREVIEWER COMMENTS

Reviewer #1 (Remarks to the Author):

The authors reported two kinds of nucleic-acid-base cocrystals, which exhibited different colors of phosphorescence from their monomeric counterparts and showed a high-temperature phosphorescence. Although the authors claimed to provide in-depth insights into the structure-properties relationship, I do not think the investigations shown here are solid enough to draw a full picture of how the hydrogen bonding network stabilizes the long-lived triplet state and contributes to the high-temperature phosphorescence. Besides, in terms of their potential applications in photodynamic therapy and multi-level information security, I am afraid there are no exciting advances that have been made by these two cocrystals. Therefore, I do not recommend it for publication in Nature Communications.

Here are some observations and inquiries:

1. It appears that the strong hydrogen bonding network plays an essential role in the design of the as-proposed nucleic-acid-base cocrystals, but from the introduction section, the rationale of incorporating such a strong hydrogen bond network is not clear. Why it can suppress the non-radiative transitions and stabilize the triplet state?
2. Please add more description on Figure 1b. It is difficult to understand what was exactly happening in the Figure.
3. In line 89, page 4, the authors claim that "From delayed photoluminescence spectra, U-MA shows an afterglow emission at 440 nm while U-B is at 540 nm (Fig.1a)". However, I do not see the data to support such statements.
4. In line 90, page 4, the authors claim that U shows different kinetics at 450 nm and 500 nm, but subsequently a lifetime of 76 ms is extracted for U and put into comparison with U-MA and U-B. Please specify what are those lifetimes exactly refer to? Besides, in this paragraph, Fig. 1a-d are all mislabeled.
5. In line 96, page 4, It is too early to conclude that strong hydrogen bonding interactions greatly enhance the phosphorescence performance, and strong hydrogen bonding networks can stabilize the triplet state. Since at this stage, the strong hydrogen bonding of the crystals in a condensed solid state remains to be validated, and the link between the strong hydrogen bond and these photoluminescent properties is not directly elucidated.
6. How the strong hydrogen bonding leads to long-lived triplets? Besides, the high temperature would affect the hydrogen bonding and then affect the photoluminescence. Yet, the investigations in this regard are missing.

7. In terms of potential applications, please justify the particular advantage of these two cocrystals or the challenges that have been conquered by these two cocrystals.

Reviewer #2 (Remarks to the Author):

In this work, the authors reported two photofunctional cocrystals with high temperature phosphorescence and photodynamic effect. This work provides a simple and effective strategy to design novel RTP materials by constructing rigid hydrogen network. The quantum yield and lifetime of phosphorescence cocrystals have been significantly enhanced, and the phosphorescent color can be manipulated. Further, the two cocrystal exhibit microbicidal effects under light. It can kill bacteria on a tooth surface and significantly helps prevent dental caries. This result does bring novelty and significance to the field. Also, this paper is generally well-written and sufficient data and analysis are provided. Therefore, I think it is worthy of publication in the Nature Communications after some minor revisions, with the details as below:

1. Introduction: the basic definition and wide applications of cocrystals need to be described. The authors need to check and include pioneering reviews in this field (Mater. Horiz. 2014, 1, 46; Chem. Eur. J. 2015, 21, 4880).
2. It is an interesting phenomenon that only one obvious RTP peak can be observed in two cocrystals but show wide emission in uracil. Please explain why?
3. Does the ratio of monomers affect the structures and related phosphorescence properties of the cocrystals?
4. The authors state two crystals show phosphorescence even in high temperature. After experiencing high temperature, will the properties of the cocrystals change?
5. The authors can explain more about Fig 5c (Micro-CT images of dentin with different treatment) to make it easier to understand.
6. The photofunctional cocrystals and RTP materials are hot topics in chemistry and materials. To arouse a broad interest from readership in this field, some strongly related works on fabrication of luminescent cocrystals (ACS Central Science, 2020, 6, 1169; J. Mater. Chem. C 2016, 4, 2527) and RTP materials for information security (Nat. Commun. 2023, 14, 1654; Angew. Chem. Int. Ed. 2023, e202302751; Sci. Bull. 2022, 67, 2076) could be added as references.
7. Line 81: "theae" should be "these". Line 137: "a very strong hydrogen bonds" should be "very strong hydrogen bonds".

Reviewer #3 (Remarks to the Author):

Organic eutectic materials have efficient light absorption and excellent photothermal conversion efficiency. The researchers chose uracil (U) as the hydrogen acceptor and melamine (MA) or boric acid (B) as the hydrogen donor and utilized strong hydrogen bonding to form two kinds of nucleic acid-based photo-functional co-crystals, which were constructed in the form of complementary base pairing. The antimicrobial properties were verified by in vitro experiments. The design idea of this manuscript is unique. However, the experimental design and validation are not detailed enough, and specific problems need more improvement and supplementation.

1. The manuscript lacks relevant characterization of the crystal structure. Please add scanning electron microscopy (SEM) images of single crystals and co-crystals and molecular orbital diagrams.
2. How were the sample ratios determined for preparing crystals of U-MA and U-B? Is there a better ratio?
3. The manuscript lacks sufficient data to show that this co-crystal achieves higher phosphorescence quantum yield and lifetime. Please add relevant characterization.
4. The manuscript mentions that U-MA forms a unique packing pattern. Please add packing images of its crystals.
5. 365nm 10W UV light has good sterilization performance. Why is there no significant difference in microbial survival between the control group and those treated with blue light irradiation, U-MA, or U-B, as mentioned in the manuscript?
6. Lacking sufficient data to prove that the antimicrobial properties of this material are provided by photoexcitation. Please add relevant characterizations—for example, fluorescent probe DCFH-DA for reactive oxygen species detection.
7. What is the minimum inhibitory concentration? How was it determined? Not mentioned in the manuscript, please add.
8. How cytocompatibility is U-MA or U-B under visible light excitation? Please add.
9. Please add characterization related to $1O_2$ detection, such as single-linear oxygen fluorescent probes (SOSG, ABDA, DPBF, and TEMP).
10. Simulating the process of human diseases in animal models helps to gain a deeper understanding of the pathogenesis of the diseases and provides a scientific basis for the biomedical application of this material. Additional in vivo experiments are recommended.

Responses to Reviewers' Comments

Journal: *Nature Communications* (NCOMMS-23-17078)

Title: Nucleic-Acid-Base Photofunctional Cocrystal for Information Security and Antimicrobial Applications

Reviewer #1

The authors reported two kinds of nucleic-acid-base cocrystals, which exhibited different colors of phosphorescence from their monomeric counterparts and showed a high-temperature phosphorescence. Although the authors claimed to provide in-depth insights into the structure-properties relationship, I do not think the investigations shown here are solid enough to draw a full picture of how the hydrogen bonding network stabilizes the long-lived triplet state and contributes to the high-temperature phosphorescence. Besides, in terms of their potential applications in photodynamic therapy and multi-level information security, I am afraid there are no exciting advances that have been made by these two cocrystals. Therefore, I do not recommend it for publication in Nature Communications. Here are some observations and inquiries:

1.It appears that the strong hydrogen bonding network plays an essential role in the design of the as-proposed nucleic-acid-base cocrystals, but from the introduction section, the rationale of incorporating such a strong hydrogen bond network is not clear. Why it can suppress the non-radiative transitions and stabilize the triplet state?

Author's Response: Thank you very much for your advice, we revised the introduction of the manuscript and added the explanation and a relevant reference (in Lines 65-68) about how the hydrogen bond network can suppress non-radiative transitions and stabilize the triplet state.

2.Please add more description on Figure 1b. It is difficult to understand what was exactly happening in the Figure.

Author's Response: Thank you very much for your suggestion. We added more necessary descriptions to Figure 1b, and the revised Figure 1 in the manuscript.

Fig. 1 Schematic illustration of the ucleic-acid-base photofunctional cocrystal.

3. In line 89, page 4, the authors claim that “From delayed photoluminescence spectra, U-MA shows an afterglow emission at 440 nm while U-B is at 540 nm (Fig. 1a)”. However, I do not see the data to support such statements.

Author’s Response: Thank you for your reminder. We are very sorry that we mistakenly placed the steady-state spectrum instead of the delayed photoluminescence spectrum. The delayed photoluminescence spectra are shown in Supplementary Fig. 6 as also shown below. We revised the content in the manuscript and replaced the relevant spectrum in the Supplementary information.

Supplementary Fig 6. The normalized delay photoluminescence spectrum (5 ms delay) of U, U-MA, and U-B powders upon 365 nm excitation.

4. In line 90, page 4, the authors claim that U shows different kinetics at 450 nm and 500 nm, but subsequently a lifetime of 76 ms is extracted for U and put into comparison with U-MA and U-B. Please specify what are those lifetimes exactly refer to? Besides, in this paragraph, Fig. 1a-d are all mislabeled.

Author’s Response: Thank you for your suggestion. We have added the kinetics comparison of the 450 nm and 500 nm luminescence peaks of U at two temperatures of 250K and 298K respectively (Supplementary Fig. 7). From the kinetics diagram, it can be seen that the slopes of the two kinetics curves are obviously different, indicating that the two phosphorescent decay channels are different. This result shows that the two luminescence peaks come from different excited state energy levels and have different lifetimes. In addition, we specify the lifetime refer to which compound and emission peak. The lifetime of U is only 76 ms at 500 nm under ambient conditions, while the lifetime of U-MA at 450 nm and U-B at 500 nm is 530 ms and 663 ms, which increase by nearly 10 times compared to U. We have revised the content in the manuscript in Lines 104-106 and rechecked and corrected all incorrect labels.

Supplementary Fig 7. The comparison of the phosphorescent kinetics decay of U at 450 nm and 500 nm at (a) 250 K. (b) 298 K.

5. In line 96, page 4, It is too early to conclude that strong hydrogen bonding interactions greatly enhance the phosphorescence performance, and strong hydrogen bonding networks can stabilize the triplet state. Since at this stage, the strong hydrogen bonding of the crystals in a condensed solid state remains to be validated, and the link between the strong hydrogen bond and these photoluminescent properties is not directly elucidated.

Author's Response: Thank you very much for your comment. We have revised the content of the manuscript and we have added SEM, single crystal structures, and computational simulation results (see Supplementary Figures 1, 2, 3, 4, 12, 13 below and also in the Supplementary Information) to demonstrate the existence of a very strong hydrogen bonding network in the cocrystal in Lines 82-89, highlighted in yellow. In the photophysical characterization section, a comparison of the phosphorescence lifetimes and quantum yield of the cocrystals and the monomer was added. This strong hydrogen bond network can increase the phosphorescence lifetime and quantum yield of the cocrystals, see Lines 106-112, highlighted in yellow.

(a)

(b)

(c)

(d)

Supplementary Fig 1. The SEM images of U-MA.

Supplementary Fig 2. The SEM images of U-B.

Supplementary Fig 3. The intermolecular hydrogen bond mode of U-MA single crystal structure.

Supplementary Fig 4. The simulated intermolecular hydrogen bond mode of U-B.

Supplementary Fig 12. The comparison of phosphorescent kinetic decay of (a) U and U-MA at 450 nm (b) U and U-B at 500 nm.

Supplementary Fig 13. The absolute quantum yield and phosphorescence quantum yield of (a) U. (b) U-MA. (c) U-B.

6. How the strong hydrogen bonding leads to long-lived triplets? Besides, the high temperature would affect the hydrogen bonding and then affect the photoluminescence. Yet, the investigations in this regard are missing.

Author's Response: After the luminescent molecule is excited by light, it generates a triplet state through intersystem crossing. The lifetime of the triplet state is generally on the level of microseconds, and it returns to the ground state mainly through non-radiative transitions (vibration, heat) at room temperature. The triplet state is sensitive to temperature and readily decays by motion so most phosphorescence only can be observed in 77K. However, in the rigid hydrogen bond network, the luminescent molecules are reasonably connected by hydrogen bonds, and the molecular motion is greatly restricted, resulting in the strong suppression of the non-radiative transitions, thus the lifetime and quantum yield of the triplet state is significantly enhanced. Consequently, the long-lived triplet state returns to the ground state via a radiative transition (phosphorescence) (see for example Adv. Mater. 28, 9920-9940, 2016, Small 18, 2104073, 2022, Nat. Rev. Mater 5, 869-885, 2020). In Fig. 2e photographs are shown of the emission of two cocrystals at different temperatures. From Fig. 2e, it can be straightforwardly observed that the emission color and duration of the two cocrystals change with temperature. As the temperature rises, the afterglow time shortens but the color of the light does not change as observed by the naked eye. Supplementary Fig 10 and 11 show the dynamic decay curves of two cocrystals at different temperatures. This shows the lifetime decay of the afterglow as the temperature rises, but it is still within the lifetime range of the long afterglow (above 100 ms). Fig 2b and 2c show the phosphorescence spectrum of two cocrystals at different temperatures. From this figure, the emission peak does not move with the rise of the temperature, which indicates that the luminescence color of the cocrystal does not change in this case when the temperature rises.

Fig 3 Phosphorescence spectra of the (b) U-MA cocrystal (c) U-B cocrystal under vacuum conditions at different temperatures.

Supplementary Fig 10. The phosphorescent kinetic of U-MA under vacuum conditions at different temperatures **(a)** at 443 nm. **(b)** at 460 nm.

Supplementary Fig 11. The phosphorescent kinetic of U-B under vacuum conditions at different temperatures **(a)** at 453 nm. **(b)** at 536 nm.

7. In terms of potential applications, please justify the particular advantage of these two cocrystals or the challenges that have been conquered by these two cocrystals.

Author's Response: As the temperature rises, the molecular thermal motion intensifies, and the non-radiative transition of the excited state will be enhanced, while the triplet state has a longer lifetime and is more easily affected by temperature, so the phosphorescence will be quenched by high temperature. Generally, the phosphorescence of small organic molecules is usually observed in 77 K, and most of the currently developed phosphorescent molecules can only be observed at room temperature, and there are few reports on materials with high-temperature phosphorescence. Therefore, most phosphorescent materials can only be used for anti-counterfeiting, imaging, coding and other applications at room temperature. Based on the high-temperature phosphorescence characteristics of cocrystal materials, it is possible to develop temperature-responsive luminescent materials, high-temperature-resistant anti-counterfeiting materials, and multiple information encryption through temperature control. Moreover, because the cocrystal can generate a large

number of long-lived triplet states under light irradiation, it can undergo energy transfer with the surrounding oxygen to generate reactive oxygen species. Our in vitro experiments also demonstrated that the two cocrystals have a spectrum of bactericidal effects, which can kill bacteria and fungi. We further conducted experiments on tooth blocks and found that the cocrystal can kill bacteria attached to teeth and has potential anti-caries applications. To the best of our knowledge, this is the first case reported application of a material with phosphorescent properties in sterilization and anti-caries in dentistry. In the future, we will attempt to develop more multifunctional cocrystal materials, which can not only be used in mature phosphorescent materials such as anti-counterfeiting and imaging, but also in new directions such as anti-caries, tooth whitening, and wound healing.

Reviewer #2

In this work, the authors reported two photofunctional cocrystals with high temperature phosphorescence and photodynamic effect. This work provides a simple and effective strategy to design novel RTP materials by constructing rigid hydrogen network. The quantum yield and lifetime of phosphorescence cocrystals have been significantly enhanced, and the phosphorescent color can be manipulated. Further, the two cocrystal exhibit microbicidal effects under light. It can kill bacteria on a tooth surface and significantly helps prevent dental caries. This result does bring novelty and significance to the field. Also, this paper is generally well-written and sufficient data and analysis are provided. Therefore, I think it is worthy of publication in the Nature Communications after some minor revisions, with the details as below:

1. Introduction: the basic definition and wide applications of cocrystals need to be described. The authors need to check and include pioneering reviews in this field (Mater. Horiz. 2014, 1, 46; Chem. Eur. J. 2015, 21, 4880).

Author's Response: Thank you very much for your suggestion. These reviews are strongly related to our work, so we have cited these publications as references 6-7, highlighted in yellow.

the

2. It is an interesting phenomenon that only one obvious RTP peak can be observed in two cocrystals but show wide emission in uracil. Please explain why?

Author's Response: From the comparison of the phosphorescent kinetics decay of U at 450 nm and 500 nm, kinetic curves exhibit different slopes indicating that these two emission peaks are from different excited state decay channels. This shows that the phosphorescence spectrum of U is composed of two parts, showing both single molecule phosphorescence and aggregate phosphorescence. Due to the unique stacking structure of U-MA, each U appears to be completely isolated within the crystal lattice, with no interaction between U. Therefore, U-MA exhibits the characteristics of single-molecule phosphorescence. For U-B, the interactions and π - π stacking between U still exist. Due to the small molecular structure of boric acid, although it can form very strong hydrogen bonds with U, it is not capable of completely isolating the phosphorophores, resulting in U-B exhibiting aggregated phosphorescence.

3. *Does the ratio of monomers affect the structures and related phosphorescence properties of the cocrystals?*

Author's Response: Thank you for your question. We have tried different ratios of monomers such as 1:1, 1:2 1:3 in mole ratio to prepare two cocrystals. From the single-crystal structure characterization, different molar ratios of monomers can only yield one type of crystal. Moreover, cocrystals cultivated in different proportions exhibit the same phosphorescence characteristics. Therefore, the ratio of monomers seems not to affect the structures and related phosphorescence properties of the cocrystals over the ratios examined.

4. *The authors state two crystals show phosphorescence even in high temperature. After experiencing high temperature, will the properties of the cocrystals change?*

Author's Response: Thank you very much for the question. The cocrystal has excellent heat resistance due to the strong hydrogen bond network. After experiencing high-temperature process, the cocrystals remain the same phosphorescent properties including afterglow color and lifetime.

5. *The authors can explain more about Fig 5c (Micro-CT images of dentin with different treatment) to make it easier to understand.*

Author's Response: Thank you very much for your suggestion. We added the explanation of the Micro-CT images in Lines 218-219, highlighted in yellow.

6. *The photofunctional cocrystals and RTP materials are hot topics in chemistry and materials. To arouse a broad interest from readership in this field, some strongly related works on fabrication of luminescent cocrystals (ACS Central Science, 2020, 6, 1169; J. Mater. Chem. C 2016, 4, 2527) and RTP materials for information security (Nat. Commun. 2023, 14, 1654; Angew. Chem. Int. Ed. 2023, e202302751; Sci. Bull. 2022, 67, 2076) could be added as references.*

Author's Response: Thank you very much for your suggestion. These publications are strongly related to our work, so we have cited these publications as references 8-9 and 48-50, highlighted in yellow.

7. *Line 81: "theae" should be "these". Line 137: "a very strong hydrogen bonds" should be "very strong hydrogen bonds".*

Author's Response: Thanks for your correction, we corrected them in the manuscript.

Reviewer #3

Organic eutectic materials have efficient light absorption and excellent photothermal conversion efficiency. The researchers chose uracil (U) as the hydrogen acceptor and melamine (MA) or boric acid (B) as the hydrogen donor and utilized strong hydrogen bonding to form two kinds of nucleic acid-based photo-functional co-crystals, which were constructed in the form of complementary base pairing. The antimicrobial properties were verified by in vitro experiments. The design idea of this manuscript is unique. However, the experimental design and validation are not detailed enough, and specific problems need more improvement and supplementation.

. The manuscript lacks relevant characterization of the crystal structure. Please add scanning electron microscopy (SEM) images of single crystals and co-crystals and molecular orbital diagrams.

Author's Response: Thank you very much for your suggestions, we added the scanning electron microscopy images of U-MA and U-B in Supplementary Fig. 1 and 2. And molecular orbital diagrams in Supplementary Fig. 20-22.

Supplementary Fig 1. The SEM image of U-MA.

(a)

(b)

(c)

(d)

Supplementary Fig 2. The SEM image of U-B.

Supplementary Fig 20. The molecular orbital of U.

Supplementary Fig 21. The molecular orbital of U-MA.

Supplementary Fig 22. The molecular orbital of U-B.

2. How were the sample ratios determined for preparing crystals of U-MA and U-B? Is there a better ratio?

Author's Response: Thanks for your comment. We have tried various ratios to prepare the cocrystal, such as molar ratios of 1:1, 1:2, 1:3 and other different ratios. Through the characterization of the single crystal structure, we found that different molar ratios will not affect the structure of the crystal and exhibit the same packing mode. Therefore, we chose the molar ratio of 1:1 to prepare the cocrystal in this work. The reason may be that the cocrystal is constructed through hydrogen bond network interaction, and the sites where the hydrogen bond donor and acceptor act are relatively fixed, so only one thermodynamically stable crystal form can be formed. Based on your comment, we revised the manuscript and added this preparation information in Line 243-245, highlighted in yellow.

3. The manuscript lacks sufficient data to show that this co-crystal achieves higher phosphorescence quantum yield and lifetime. Please add relevant characterization.

Author's Response: Thank you very much for your suggestions, we add the comparison of phosphorescent properties (including lifetime and quantum yield) of the U monomer and two cocrystals (U-MA and U-B) in Supplementary Fig.12 and 13 and highlight this data in lines 105-112 of the manuscript.

Supplementary Fig 12. The comparison of phosphorescent kinetic decay of (a) U and U-MA at 450 nm (b) U and U-B at 500 nm.

Supplementary Fig 13. The absolute quantum yield and phosphorescence quantum yield of (a) U. (b) U-MA. (c) U-B.

4. The manuscript mentions that U-MA forms a unique packing pattern. Please add packing images of its crystals.

Author's Response: Thank you for your comment. The single crystal structure is shown in Figure 3e. The single crystal structure shows the unique packing mode of the U-MA cocrystal. In the same layer, U forms a base-pairing hydrogen bond with MA leading to no interaction with U each other. In the different layer, U has no interaction with U in the other layer. Based on your comment, we revised the manuscript and added this preparation information in Lines 138-140, highlighted in yellow.

Figure 3e. The unique packing mode of the U-MA cocrystal.

5. 365nm 10W UV light has good sterilization performance. Why is there no significant difference in microbial survival between the control group and those treated with blue light irradiation, U-MA, or U-B, as mentioned in the manuscript?

Author's Response: Thank you for your comment. On the one hand, due to the structure of the UV lamp, the light is relatively divergent. We do not use a lens to focus, so the actual power irradiated to the sample will be much smaller than 10 W. On the other hand, in order to eliminate the thermal effect caused by long-term exposure to the 10W UV lamp and the possible interference of the sterilization effect of the UV lamp, we increased the distance between the UV lamp and the bacteria. In our sterilization experiments, the distance between the UV lamp and the culture medium was approximately 50 cm. On the premise of ensuring that the UV lamp can irritate the culture medium, we try to increase the distance between the UV lamp and the culture medium as much as possible. Based on the above two reasons, the impact of the sterilization effect of ultraviolet light on the experiment was avoided as much as possible. Based on your comment, we revised the manuscript and added this preparation information in Lines 265-266, highlighted in yellow.

6. Lacking sufficient data to prove that the antimicrobial properties of this material are provided by photoexcitation. Please add relevant characterizations—for example, fluorescent probe DCFH-DA for reactive oxygen species detection.

Author's Response: Thank you for your comment. First of all, we conducted the control experiment, including a control group, only light irradiation, only cocrystal and cocrystal under photoexcitation. Through different characterization including confocal microscopy, only two cocrystals under photoexcitation show the antimicrobial properties. Furthermore, we use the SOSG probe to detect the reactive oxygen species (Supplementary Fig 27). The fluorescence response of SOSG at 525 nm indicates that the generation of reactive oxygen species of two cocrystals under photoexcitation is singlet oxygen. Therefore, these results prove that these two cocrystals have antimicrobial properties

under photoexcitation. Based on your comment, we revised this content in the manuscript in Lines 177-179, highlighted in yellow.

Supplementary Fig 27. Fluorescence response of SOSG upon treatment with (a) U-MA and (b) U-B under excitation at 365 nm for singlet oxygen generation, $\lambda_{ex} = 504$ nm.

7. What is the minimum inhibitory concentration? How was it determined? Not mentioned in the manuscript, please add.

Author's Response: Thank you very much for your suggestion. We added the minimum inhibitory concentration test. The method is shown in the manuscript in Lines 264-269 and the result is shown in Supplementary Table 1, highlighted in yellow.

8. How cytocompatibility is U-MA or U-B under visible light excitation? Please add.

Author's Response: Thank you for your comment. The cytocompatibility of U-MA or U-B is shown in Supplementary Figure 32. The experiment was conducted under normal experimental conditions with light shielding, that is, under visible light conditions. Since visible light cannot excite the eutectic material to generate reactive oxygen species, the cytotoxicity is low.

Supplementary Fig 32. Cell viability of HGF cells under treatment with different concentration of U-MA and U-B. Error bars = Standard Deviation (n=3).

9. Please add characterization related to 1O_2 detection, such as single-linear oxygen fluorescent probes (SOSG, ABDA, DPBF, and TEMP).

Author's Response: Thank you for your comment. We used the singlet oxygen probe SOSG to detect reactive oxygen species in two cocrystals U-MA and U-B. The fluorescence response of SOSG upon treatment with U-MA and U-B under excitation at 365 nm is shown in Supplementary Fig 27. With the increased irradiation time of the cocrystal, the intensity of fluorescence peak at 525 nm increases indicating that the fluorescence of SOSG is opened. This result demonstrates that the generation of reactive oxygen species of these two cocrystals is singlet oxygen. Based on your comment, we revised this content in the manuscript in Lines 177-179, highlighted in yellow.

Supplementary Fig 27. Fluorescence response of SOSG upon treatment with (a) U-MA and (b) U-B under excitation at 365 nm for singlet oxygen generation, $\lambda_{ex} = 504$ nm.

10. Simulating the process of human diseases in animal models helps to gain a deeper understanding of the pathogenesis of the diseases and provides a scientific basis for the biomedical application of this material. Additional *in vivo* experiments are recommended.

Author's Response: Thank you very much for your suggestion, simulating the process of human diseases in animal models can help us have a better understanding of the use of medical materials in human diseases. However, animal research in Hong Kong requires complex, rigorous and time-consuming ethical applications, and routine animal testing for caries prevention takes 2-3 months to complete, so it is difficult for us to increase *in vivo* testing at this stage. In this research, we are proposing the potential application of phosphorescent materials, which can indeed sterilize the tooth surface, reduce *Streptococcus mutans*, and then initially simulate its use in caries management. A large number of caries prevention animal experiments (Lei L, Zhang Y, Xu Y, Tian Y, Zhao J, Xiang Y, Yang H, Yang Y, Hu T. Spermine-starch nanoparticles with antisense vicR suppress *Streptococcus mutans* cariogenicity. *J Mater Chem B*. 2023 Jun 28;11(25):5752-5766; Chen H, Tang Y, Weir MD, Gao J, Imazato S, Oates TW, Lei L, Wang S, Hu T, Xu HHK. Effects of *S. mutans* gene-modification and antibacterial monomer dimethylaminohexadecyl methacrylate on biofilm growth and acid

production. Dent Mater. 2020 Feb;36(2):296-309) completed by the corresponding author of the project, Professor Hu Tao, and his international colleagues, have shown that reducing and intervening *Streptococcus mutans* has a significant effect on arresting dental caries *in vivo*. In the future research plan, we will try to add vivo experiments, deepen our understanding of the pathogenesis of the diseases, and expand the scope of application of the cocrystal phosphorescent materials in cariology and other areas of medical application.

REVIEWER COMMENTS

Reviewer #1 (Remarks to the Author):

I have carefully reviewed the revised manuscript entitled “Nucleic-Acid-Base Photofunctional Cocrystal for Information Security and Antimicrobial Applications” and the authors' responses to the reviewers' comments. The authors have thoroughly addressed most of the concerns raised previously. The concept of using strong hydrogen bonding to construct photofunctional cocrystals with enhanced phosphorescence performance and antimicrobial effects is interesting. The additional data provided, especially on demonstrating the hydrogen bonding networks and comparison of the phosphorescent properties between the cocrystals and monomer, strengthen the conclusions. However, a few issues need to be further addressed before final acceptance.

1. The authors claim unique packing in the U-MA cocrystal contributes to the blue-shifted single molecule phosphorescence, but no direct evidence definitively proves this. Further computational modeling or spectroscopic analysis that can simulate the effect of packing mode is still needed.
2. For the application in information security, please discuss if the phosphorescence lifetime, intensity or color stability over temperature cycling has been tested. This is important for practical usage through temperature changes.
3. For antimicrobial effects, please discuss the ROS generation capacity and efficiency in comparison to other existing antimicrobial photocatalysts/agents. This would better highlight the advantage.
4. Specify in the methods the cell type and assays used for the biocompatibility evaluation.

Reviewer #2 (Remarks to the Author):

In my view, the authors have answered all the questions and revised related points, and thus this revised work can be published as it is.

Reviewer #3 (Remarks to the Author):

The revised manuscript is ready for publication.

Responses to the Reviewer's Comments

Journal: Nature Communications (NCOMMS-23-17078A).

Title: Nucleic-Acid-Base Photofunctional Cocrystal for Information Security and Antimicrobial Applications

Reviewer #1 (Remarks to the Author):

I have carefully reviewed the revised manuscript entitled "Nucleic-Acid-Base Photofunctional Cocrystal for Information Security and Antimicrobial Applications" and the authors' responses to the reviewers' comments. The authors have thoroughly addressed most of the concerns raised previously. The concept of using strong hydrogen bonding to construct photofunctional cocrystals with enhanced phosphorescence performance and antimicrobial effects is interesting. The additional data provided, especially on demonstrating the hydrogen bonding networks and comparison of the phosphorescent properties between the cocrystals and monomer, strengthen the conclusions. However, a few issues need to be further addressed before final acceptance.

1. The authors claim unique packing in the U-MA cocrystal contributes to the blue-shifted single molecule phosphorescence, but no direct evidence definitively proves this. Further computational modeling or spectroscopic analysis that can simulate the effect of packing mode is still needed.

Author's Response: Thank you very much for your suggestion. We tested the phosphorescence spectra of dilute uracil solution at 77 K (Supplementary Fig 17). Since the concentration of the solution is about 10^{-4} - 10^{-5} , uracil is surrounded by solvent molecules, and there is no interaction between uracil, so it shows the single molecule phosphorescence spectrum of uracil. Therefore, by comparing the phosphorescence spectrum of U-MA and the low-temperature phosphorescence spectrum of U dilute solution, it is found that the two spectra show a good coincidence. The fs-transient absorption spectrum of U-MA and U monomer film exhibit similar evolution in the excited state (Fig 3 abc and Supplementary Fig 15). These results indicate that U-MA phosphorescence is from the single molecule phosphorescence. In addition, we also carried out computer simulations. Since the computational method (mo62x) underestimates the energy of the triplet state, the calculated spectra will be redshifted compared with the actual spectra (Supplementary Fig 18). Therefore, the calculated results are in good agreement with the phosphorescent spectra of the U single molecule. The results indicate that the blue shift of phosphorescence of U-MA is due to the U-MA exhibit single molecule phosphorescence.

Supplementary Fig 17. The phosphorescence spectrum of U at 77 K in diluted methanol solution.

Supplementary Fig 15. fs-TA spectra of U film under 290 nm excitation (a) from 0.15 to 1.27 ps. (b) from 1.27 to 2.32 ps. (c) from 2.32 ps to 6.4 ns (d) Kinetic fitting for U at 484 nm.

Fig 3. fs-TA spectra of U-MA film under 290 nm excitation.

Supplementary Fig 18. The simulated phosphorescence spectrum of U monomer.

2. For the application in information security, please discuss if the phosphorescence lifetime, intensity or color stability over temperature cycling has been tested. This is important for practical usage through temperature changes.

Author's Response: Thank you very much for your suggestion. We tested the phosphorescence color and intensity of two cocrystals at ambient temperature and 100°C cycling. It can be seen from the figure that the phosphorescent color of the two cocrystals did not change during the five heating and cooling cycles. The graph shows the phosphorescent intensity of the two cocrystals at 450 nm (U-MA) and 550 nm (U-B) as a function of cycling. From the above two results, it can be seen that these two cocrystals have very good thermal stability.

Supplementary Fig 33. Photographs of the phosphorescence of two cocrystals at room temperature and 100°C cycle.

Supplementary Fig 34. The stability of two cocrystals over the temperature cycling of (a) 450 nm of U-MA, (b) 550 nm of U-B.

3. For antimicrobial effects, please discuss the ROS generation capacity and efficiency in comparison to other existing antimicrobial photocatalysts/agents. This would better highlight the advantage.

Author's Response: Thank you very much for your suggestion. We have studied that U-MA and U-B can produce singlet oxygen through the Singlet Oxygen Sensor Green (SOSG) probe experiment in the manuscript (Supplementary Fig. 28). However, because our research focuses on whether the antibacterial activity can be achieved through reactive oxygen species, and there are limitations in reagents and instruments, we are temporarily unable to conduct tests on the singlet oxygen yield. From *in vitro* experiments, we can see that our materials have strong antibacterial effects, which shows that the singlet oxygen yield of the materials is significant.

To the best of our knowledge, it is the first case to use RTP materials in an anti-carries application. We would like to briefly introduce the applications of the new materials and hope this research can bring a new research perspective to the application of optical functional materials in the medical field, especially dental materials. Your suggestion about comparing the ROS generation capacity and efficiency with existing photocatalysts can indeed better highlight our advantages and provide ideas for our future research.

Supplementary Fig 28. Fluorescence response of SOSG upon treatment with (a) U-MA and (b) U-B under excitation at 365 nm for singlet oxygen generation, $\lambda_{ex} = 504$ nm.

4. Specify in the methods the cell type and assays used for the biocompatibility evaluation.

Author's Response: Thank you very much for your suggestion. We have introduced our cell type and assays used for the biocompatibility evaluation in the method part in Line 281-284, highlighted with yellow in the revised manuscript.

REVIEWERS' COMMENTS

Reviewer #1 (Remarks to the Author):

I think that this revised manuscript is suitable for publication without further revision.